# Minimum Eigenvector Collaborative Representation Discriminant Projection for Feature Extraction

**DOI:** 10.3390/s20174778

**Published:** 2020-08-24

**Authors:** Haoshuang Hu, Da-Zheng Feng

**Affiliations:** National Laboratory of Radar Signal Processing, Xidian University, Xi’an 710071, China; hshu@stu.xidian.edu.cn

**Keywords:** collaborative representation, discriminant projection, feature extraction, linear dimensionality reduction, subspace projection

## Abstract

High-dimensional signals, such as image signals and audio signals, usually have a sparse or low-dimensional manifold structure, which can be projected into a low-dimensional subspace to improve the efficiency and effectiveness of data processing. In this paper, we propose a linear dimensionality reduction method—minimum eigenvector collaborative representation discriminant projection—to address high-dimensional feature extraction problems. On the one hand, unlike the existing collaborative representation method, we use the eigenvector corresponding to the smallest non-zero eigenvalue of the sample covariance matrix to reduce the error of collaborative representation. On the other hand, we maintain the collaborative representation relationship of samples in the projection subspace to enhance the discriminability of the extracted features. Also, the between-class scatter of the reconstructed samples is used to improve the robustness of the projection space. The experimental results on the COIL-20 image object database, ORL, and FERET face databases, as well as Isolet database demonstrate the effectiveness of the proposed method, especially in low dimensions and small training sample size.

## 1. Introduction

High-dimensional data widely exists in real applications, such as image recognition, information retrieval, etc. Particularly, in actual data processing problems, one often encounters the so-called high-dimensiona l small sample size (SSS) problem, in which the number of available samples is smaller than the dimensionality of the sample feature. Besides, high-dimensional data contains a lot of redundant information, and directly processing high-dimensional data will consume a lot of storage and computing resources. Fortunately, some previous research work [1,2,3] has shown that high-dimensional data is likely lying on or close to a low-dimensional submanifold space, which means that high-dimensional data can be projected into a low-dimensional subspace by some dimensionality reduction (DR) methods without losing important information. In the past few decades, numerous DR theories and methods have been proposed, and part of the work can be found in [4,5,6,7,8].

Principal component analysis (PCA) [4] and linear discriminant analysis (LDA) [5] are the most classic and popular DR methods, and PCA belongs to an unsupervised method while LDA is a supervised method. Despite their simplicity and effectiveness, they still suffer from some limitations in practice. For example, PCA is an unsupervised DR method and fails to provide discrimination information for different classes of data. LDA can find at most C−1 meaningful discriminant projection directions because theoretical analysis shows that the rank of the between-class scatter matrix is at most C−1, where C represents the number of classes. More importantly, they do not use the structural information of the samples, which greatly reduces the discriminativeness of the extracted features, especially when dealing with SSS problems. Most of the DR methods proposed afterward are based on these two methods or their extended version but consider the structure of the samples. Some DR methods which focus on the local structure of the data have been developed to improve the discrimination performance of the low-dimensional space. A partial list of these methods includes the unsupervised ones such as locality preserving projections (LPP) [9], neighborhood preserving embedding (NPE) [10], and locally linear embedding (LLE) [1], and the supervised ones like marginal Fisher analysis (MFA) [11] and constrained discriminant neighborhood embedding (CDEN) [12]. Several other DR methods that utilize both the local and global structure of the data to improve recognition accuracy have been proposed, such as local Fisher discriminant analysis (LFDA) [13], locally linear discriminant embedding (LLDE) [14], and locality preserving discriminant projections (LPDP) [15].

Recently, some representation-based methods have been used for classification, including sparse representation classification (SRC) [16], sparsity preserving projections (SPP) [17], discriminant sparse neighborhood preserving embedding (DSNPE) [18], and discriminative sparsity preserving projections (DSPP) [19]. However, solving a sparse problem requires an iterative method, which is usually time-consuming. Another type of representation-based DR method, which uses L2 regularization and has a closed solution, has attracted wide attention. It has been proved in [20] that the collaborative representation classification (CRC), with higher efficiency, is competitive to SRC in terms of recognition accuracy. Since then, some collaborative representation based have been proposed, such as collaborative representation based projections (CRP) [21], regularized least square-based discriminative projections (RLSDP) [22], a collaborative representation reconstruction based projections (CRRP) [23], collaborative representation based discriminant neighborhood projections (CRDNP) [24] and collaborative preserving Fisher discriminant analysis (CPFDA) [25].

Except for the linear dimensionality reduction methods introduced above, a class of nonlinear dimensionality reduction methods has been proposed to deal with nonlinear dimensionality reduction problems. Most of the nonlinear DR methods directly use kernel trick to expand linear DR method, likes kernel PCA (KPCA) [26], kernel Fisher discriminant (KFD) [27], kernel direct discriminant analysis (KDDA) [28], and kernel collaborative representation-based projection (KCRP) [29]. Some other recent DR methods can be found in [30,31,32].

In this article, we study linear DR methods. Although some of the proposed methods make use of both the local structure and the global structure of the sample to extract features, when the reconstruction error of the sample is large, it is difficult for them to maintain the true structural similarity of the sample. In addition, most of the previous dimensionality reduction methods only consider the structural similarity of similar samples when looking for the projection subspace, while ignoring the structural similarity of different types of samples, which could also be used to improve the discriminability of the extracted features.

In order to extract the features with a strong discriminant, a minimum eigenvector collaborative representation discriminant projection (MECRDP) is proposed in this paper. The main contributions of our work are as follows. First, in the collaborative representation of samples, we not only use the information of the sample space but also consider the information of the sample eigenvector space. Specifically, we use the eigenvector corresponding to the smallest non-zero eigenvalue of the sample covariance matrix to reduce the collaborative representation error of each sample. Then, we maintain the collaborative representation relationship of the samples to improve the discriminability of the extracted features. Also, the between-class scatter of the reconstructed samples is used to improve the robustness of the projection subspace. Last, experimental results on four public databases show that MECRDP outperforms other DR methods in terms of recognition accuracy, especially in low dimensions and small training sample size.

The remainder of this paper is organized as follows. In Section 2, we briefly introduce the research work closely related to our method, including LDA and CRP. Section 3, we propose a minimum eigenvector collaborative representation discriminant projection to improve the discriminability of the projection subspace. The experimental results are presented in Section 4. Finally, the conclusion remarks are given in Section 5.

## 2. Related Works

For simplicity, suppose a training samples set of C classes is denoted by $X=[x_{1},x_{2},\ldots ,x_{n}]\in {ℜ}^{m\times n}$, where $x_{i}\in {ℜ}^{m}$ represents the ith sample, m is the dimension of the sample feature, and n is the number of samples. Besides, suppose the cth class contains $n_{c}$ samples, and $\sum_{c=1}^{C}n_{c}=n$. The method we proposed in this paper has a great relationship with LDA and CRP. In what follows, we briefly review these two methods.

### 2.1. Linear Discriminant Analysis

The goal of LDA [5] is seeking a projection matrix so that the within-class scatter is minimized, and the between-class scatter is maximized simultaneously. According to the graph embedding [13], the projection matrix of LDA corresponds to the following two optimization problems, respectively,
(1)$$\underset{P}{\max}\sum_{i,j}^{n}{\left\|P^{T}x_{i}-P^{T}x_{j}\right\|}_{2}^{2}W_{i,j}^{\left(b\right)},$$
(2)$$\underset{P}{\min}\sum_{i,j}^{n}{\left\|P^{T}x_{i}-P^{T}x_{j}\right\|}_{2}^{2}W_{i,j}^{\left(w\right)},$$
where the weights $W_{i,j}^{\left(b\right)}$ and $W_{i,j}^{\left(w\right)}$ are defined as, respectively,
(3)$$W_{i,j}^{\left(b\right)}=\{\begin{array}{cc} 1/n-1/n_{c}, & \mathrm{if}\;x_{i}\;\mathrm{and}\;x_{j}\;\mathrm{belong}\;\mathrm{to}\;\mathrm{the}\;c\mathrm{th}\;\mathrm{class},\\ 1/n, & \mathrm{otherwise}. \end{array}.$$
(4)$$W_{i,j}^{\left(w\right)}=\{\begin{array}{cc} 1/n_{c},\; & \mathrm{if}\;x_{i}\;\mathrm{and}\;x_{j}\;\mathrm{belong}\;\mathrm{to}\;\mathrm{the}\;\mathrm{same}\;\mathrm{class},\\ 0, & \mathrm{otherwise}. \end{array},$$

Using some algebraic transform, we can rewrite (1) and (2) as
(5)$$\begin{array}{cc} \underset{P}{\max}\sum_{i,j}^{n}{\left\|P^{T}x_{i}-P^{T}x_{j}\right\|}_{2}^{2}W_{i,j}^{\left(b\right)} & =tr\left(P^{T}\left(\sum_{i,j=1}^{n}(x_{i}-x_{j})W_{i,j}^{\left(b\right)}{\left(x_{i}-x_{j}\right)}^{T}\right)P\right)\\ & =tr\left(P^{T}\left(\sum_{i,j=1}^{n}\left(x_{i}W_{i,j}^{\left(b\right)}x_{i}^{T}-x_{i}W_{i,j}^{\left(b\right)}x_{j}^{T}-x_{j}W_{i,j}^{\left(b\right)}x_{i}^{T}+x_{j}W_{i,j}^{\left(b\right)}x_{j}^{T}\right)\right)P\right)\\ & =2tr\left(P^{T}\left(XD^{\left(b\right)}X^{T}-XW^{\left(b\right)}X^{T}\right)P\right)\\ & =2tr\left(P^{T}\left(X\left(D^{\left(b\right)}-W^{\left(b\right)}\right)X^{T}\right)P\right)\\ & =2tr\left(P^{T}\left(XL^{\left(b\right)}X^{T}\right)P\right)\\ & =tr\left(P^{T}S_{b}P\right) \end{array}$$
(6)$$\begin{array}{cc} \underset{P}{\min}\sum_{i,j}^{n}{\left\|P^{T}x_{i}-P^{T}x_{j}\right\|}_{2}^{2}W_{i,j}^{\left(w\right)} & =tr\left(P^{T}\left(\sum_{i,j=1}^{n}(x_{i}-x_{j})W_{i,j}^{\left(w\right)}{\left(x_{i}-x_{j}\right)}^{T}\right)P\right)\\ & =tr\left(P^{T}\left(\sum_{i,j=1}^{n}\left(x_{i}W_{i,j}^{\left(w\right)}x_{i}^{T}-x_{i}W_{i,j}^{\left(w\right)}x_{j}^{T}-x_{j}W_{i,j}^{\left(w\right)}x_{i}^{T}+x_{j}W_{i,j}^{\left(w\right)}x_{j}^{T}\right)\right)P\right)\\ & =2tr\left(P^{T}\left(XD^{\left(w\right)}X^{T}-XW^{\left(w\right)}X^{T}\right)P\right)\\ & =2tr\left(P^{T}\left(X\left(D^{\left(w\right)}-W^{\left(w\right)}\right)X^{T}\right)P\right)\\ & =2tr\left(P^{T}\left(XL^{\left(w\right)}X^{T}\right)P\right)\\ & =tr\left(P^{T}S_{w}P\right) \end{array}$$
where tr(·) denotes matrix trace, $S_{b}=2XL^{\left(b\right)}X^{T}$ and $S_{w}=2XL^{\left(w\right)}X^{T}$ are the between-class scatter matrix, and the within-class scatter matrix, respectively. $L^{\left(b\right)}=D^{\left(b\right)}-W^{\left(b\right)}$ and $L^{\left(w\right)}=D^{\left(w\right)}-W^{\left(w\right)}$ are the Laplacian matrices, in which $D^{\left(b\right)}$ and $D^{\left(w\right)}$ are diagonal matrices with their diagonal entries are $D_{i,i}^{\left(b\right)}=\sum_{j=1}^{n}W_{i,j}^{\left(b\right)}$ and $D_{i,i}^{\left(w\right)}=\sum_{j=1}^{n}W_{i,j}^{\left(w\right)}$, respectively.

Using (5) and (6), the objective function of LDA can be modeled as
(7)$$\underset{P}{\max}\frac{tr\left(P^{T}S_{b}P\right)}{tr\left(P^{T}S_{w}P\right)},$$

### 2.2. Collaborative Representation Based Projections

CRP [21] is an unsupervised discriminant projection method based on L2 regularized least squares. The collaborative representation coefficients of the ith sample is gotten by solving the following optimization problem
(8)$$\underset{r_{i}}{\min}{\left\|x_{i}-Xr_{i}\right\|}_{2}^{2}+\lambda {\left\|r_{i}\right\|}_{2}^{2},\;s.t.\;e_{i}^{T}r_{i}=0,$$
where $r_{i}\in {ℜ}^{n}$ is the collaborative representation coefficients of the ith sample, λ>0 is a regularization parameter, and $e_{i}={\left[0_{1},\ldots ,0_{i-1},1,0_{i+1},\ldots ,0_{n}\right]}^{T}$. The constraint in (1) means the ith sample is represented on all the samples other than itself. The optimal solution of (1) can be easily achieved by using the Lagrangian multiplier method as
(9)$$r_{i}=Q\left(Xx_{i}-\frac{e_{i}^{T}QXx_{i}}{e_{i}^{T}Qe_{i}}e_{i}\right),$$
where $Q={\left(X^{T}X+\lambda I\right)}^{-1}$ and $Ι\in {ℜ}^{n\times n}$ is a identity matrix.

Using the collaborative representation coefficients, the optimal projection matrix $P\in {ℜ}^{m\times d}$ (d<m) of CRP is obtained by solving the following two optimization problems, simultaneously
(10)$$\underset{P}{\min}\sum_{i=1}^{n}{\left\|P^{T}x_{i}-\sum_{j=1}^{n}r_{i,j}P^{T}x_{j}\right\|}_{2}^{2},$$

(11)$$\underset{P}{\max}\sum_{i=1}^{n}{\left\|P^{T}x_{i}-P^{T}\bar{x}\right\|}_{2}^{2},$$
where $r_{i,j}$ denotes the jth entry of $r_{i}$, and $\bar{x}=(1/n)\sum_{i=1}^{n}x_{i}$ is the mean of the samples.

With some algebraic formulations, (10) and (11) can be, respectively, rewritten as
(12)$$\begin{array}{cc} \underset{P}{\min}\sum_{i=1}^{n}{\left\|P^{T}x_{i}-\sum_{j=1}^{n}r_{i,j}P^{T}x_{j}\right\|}_{2}^{2} & =\sum_{i=1}^{n}{\left\|P^{T}Xe_{i}-P^{T}Xr_{i}\right\|}_{2}^{2}\\ & =tr\left(P^{T}X\left(\sum_{i=1}^{n}(e_{i}-r_{i}){\left(e_{i}-r_{i}\right)}^{T}\right)X^{T}P\right)\\ & =tr\left(P^{T}X(I-R){\left(I-R\right)}^{T}X^{T}P\right)\\ & =tr\left(P^{T}S_{L}P\right) \end{array}$$
(13)$$\begin{array}{cc} \underset{P}{\max}\sum_{i=1}^{n}{\left\|P^{T}x_{i}-P^{T}\bar{x}\right\|}_{2}^{2} & =tr\left(P^{T}\left(\sum_{i}^{n}(x_{i}-\bar{x}){\left(x_{i}-\bar{x}\right)}^{T}\right)P\right)\\ & =tr\left(P^{T}S_{T}P\right) \end{array}$$
where $R=[r_{1},r_{2},\ldots ,r_{n}]$ is the representation coefficients matrix, $S_{L}=(I-R){\left(I-R\right)}^{T}$ denotes the local scatter matrix and $S_{T}=\sum_{i}^{n}(x_{i}-\bar{x}){\left(x_{i}-\bar{x}\right)}^{T}$ is the total scatter matrix.

Then, the optimal projection matrix of the CRP is gotten by solving the following optimization problem
(14)$$\underset{P}{\min}\frac{tr\left(P^{T}S_{L}P\right)}{tr\left(P^{T}S_{T}P\right)}$$

## 3. Minimum Eigenvector Collaborative Representation Discriminant Projection

LDA uses between-class scatter and within-class scatter to improve the discrimination of extracted features, but it ignores the structural relationship of the samples, resulting in a decline in the discrimination of the features in the projection subspace. Although CRP takes into account the structural relationship of the sample, it does not use the class information of the sample, which is not conducive to the improvement of recognition accuracy. In particular, when the number of samples is small, there will be a large representation error, ${\left\|x_{i}-Xr_{i}\right\|}_{2}^{2}$, which fails to maintain the similarity of the sample structure. Here, we propose a new feature extraction method to alleviate the problems mentioned above.

### 3.1. Method Proposed

The sample is represented on all samples other than itself; therefore, this may cause a large reconstruction error when the number of samples is small. However, any eigenvector corresponding to a non-zero eigenvalue of the sample covariance matrix contains partial information of each sample. In order to improve the reconstruction error of the sample, but only use very little information of the represented sample, we consider constructing an expanded sample matrix $\tilde{X}$ with the eigenvector corresponding to the smallest non-zero eigenvalue of the sample covariance matrix. Let $x_{v}$ be the eigenvector corresponding to the smallest non-zero eigenvalue of $XX^{T}$, then the expanded sample matrix $\tilde{X}$ is defined as $\tilde{X}=[X,x_{v}]$. Similar to (8), the collaborative representation coefficients of the ith sample is achieved by
(15)$$\underset{{\tilde{r}}_{i}}{\min}{\left\|x_{i}-\tilde{X}{\tilde{r}}_{i}\right\|}_{2}^{2}+\lambda {\left\|{\tilde{r}}_{i}\right\|}_{2}^{2},\;s.t.\;{\tilde{e}}_{i}^{T}{\tilde{r}}_{i}=0,$$
where ${\tilde{r}}_{i}\in {ℜ}^{n+1}$ is the collaborative representation coefficient and ${\tilde{e}}_{i}={\left[e_{i}^{T},0\right]}^{T}$. Let $\tilde{R}=[{\tilde{r}}_{1},{\tilde{r}}_{2},\ldots ,{\tilde{r}}_{n}]$ be the collaborative representation coefficient matrix.

Then, the sample $x_{i}$ can be reconstructed as
(16)$${\hat{x}}_{i}=\tilde{X}{\tilde{r}}_{i},$$
and the reconstructed sample matrix is
(17)$$\hat{X}=[{\hat{x}}_{1},{\hat{x}}_{2},\ldots ,{\hat{x}}_{n}]=X[{\hat{r}}_{1},{\hat{r}}_{2},\ldots ,{\hat{r}}_{n}]=X\tilde{R}$$

In order to maintain the reconstruction similarity of the samples and keep within-class compactness, we modified the optimization problem in (2) as
(18)$$\underset{P}{\min}\sum_{i,j}^{n}{\left\|P^{T}x_{i}-P^{T}{\hat{x}}_{j}\right\|}_{2}^{2}W_{i,j}^{\left(w\right)},$$

Similar to (6), we can simplify (18) as
(19)$$\begin{array}{cc} \underset{P}{\min}\sum_{i,j}^{n}{\left\|P^{T}x_{i}-P^{T}{\hat{x}}_{j}\right\|}_{2}^{2}W_{i,j}^{\left(w\right)} & =tr\left(P^{T}\left(\sum_{i,j}^{n}\left(x_{i}-{\hat{x}}_{j}\right)W_{i,j}^{\left(w\right)}{\left(x_{i}-{\hat{x}}_{j}\right)}^{T}\right)P\right)\\ & =tr\left(P^{T}\left(\sum_{i,j}^{n}\left(x_{i}W_{i,j}^{\left(w\right)}x_{i}^{T}-{\hat{x}}_{j}W_{i,j}^{\left(w\right)}x_{i}^{T}-x_{i}W_{i,j}^{\left(w\right)}{\hat{x}}_{j}^{T}+{\hat{x}}_{j}W_{i,j}^{\left(w\right)}{\hat{x}}_{j}^{T}\right)\right)P\right)\\ & =tr\left(P^{T}\left(XD^{\left(w\right)}X^{T}-\hat{X}W^{\left(w\right)}X^{T}-XW^{\left(w\right)}{\hat{X}}^{T}+\hat{X}D^{\left(w\right)}{\hat{X}}^{T}\right)P\right)\\ & =tr\left(P^{T}{\overset{⌢}{S}}_{w}P\right) \end{array}$$
where ${\overset{⌢}{S}}_{w}=XD^{\left(w\right)}X^{T}-\hat{X}W^{\left(w\right)}X^{T}-XW^{\left(w\right)}{\hat{X}}^{T}+\hat{X}D^{\left(w\right)}{\hat{X}}^{T}$ is the collaborative within-class scatter matrix and $W^{\left(w\right)}$*,*$D^{\left(w\right)}$ are defined in (4) and (6).

In what follows, we define the collaborative reconstructed between-class scatter matrix as
(20)$$\begin{array}{cc} {\hat{S}}_{b} & =\sum_{i,j}^{n}\left({\hat{x}}_{i}-{\hat{x}}_{j}\right)W_{i,j}^{\left(b\right)}{\left({\hat{x}}_{i}-{\hat{x}}_{j}\right)}^{T}\\ & =\sum_{i,j}^{n}\left({\hat{x}}_{i}W_{i,j}^{\left(b\right)}{\hat{x}}_{i}^{T}-{\hat{x}}_{j}W_{i,j}^{\left(b\right)}{\hat{x}}_{i}^{T}-{\hat{x}}_{i}W_{i,j}^{\left(b\right)}{\hat{x}}_{j}^{T}+{\hat{x}}_{j}W_{i,j}^{\left(b\right)}{\hat{x}}_{j}^{T}\right)\\ & =\hat{X}D^{\left(b\right)}{\hat{X}}^{T}-\hat{X}W^{\left(b\right)}{\hat{X}}^{T}-\hat{X}W^{\left(b\right)}{\hat{X}}^{T}+\hat{X}D^{\left(b\right)}{\hat{X}}^{T}\\ & =2\hat{X}(D^{\left(b\right)}-W^{\left(b\right)}){\hat{X}}^{T}\\ & =2\hat{X}L^{\left(b\right)}{\hat{X}}^{T} \end{array}$$
where $L^{\left(b\right)}$ has been defined (5).

Based on the Fisher criterion, considering both the between-class scatter matrix $S_{b}$ and the collaborative reconstructed between-class scatter matrix ${\hat{S}}_{b}$, the objective function of the proposed MECRDP is formulated as
(21)$$\underset{P}{\min}\frac{tr\left(P^{T}{\overset{⌢}{S}}_{w}P\right)}{tr\left(P^{T}{\tilde{S}}_{b}P\right)},$$
where ${\tilde{S}}_{b}=\alpha S_{b}+(1-\alpha ){\hat{S}}_{b}$ and α∈[0,1] is a balance factor between $S_{b}$ and ${\hat{S}}_{b}$. According to the definition of $S_{b}$, we know that maximizing $tr(P^{T}S_{b}P)$ can improve the discrimination of the projection matrix P. However, when the dimension of the subspace exceeds a certain size, as the dimension of the projection subspace increases, the discriminant performance is more affected by noise. Consider that the collaborative representation between samples could better describe the similarity of samples in the structure, thereby reducing the impact of the noise, then we can use ${\hat{S}}_{b}$ to improve the robustness of the projection matrix.

The problem (21) is a generalized eigenvalue problem, whose optimal solution could be achieved by the generalized eigenvalue decomposition as follows
(22)$${\overset{⌢}{S}}_{w}p_{i}=\lambda_{i}{\tilde{S}}_{b}p_{i},$$
where $\lambda_{i}$ and $p_{i}$ are the eigenvalue and corresponding eigenvector, respectively. Then the projection matrix P is composed of the eigenvectors corresponding to d—the smallest non-zero eigenvalues—that is $P=[p_{1},p_{2},\ldots ,p_{d}]$.

Using $x_{v}$, different from CRP, we not only consider the information in the sample space but also use part of the information in the feature value space, so we can get a smaller representation error, ${\left\|x_{i}-Xr_{i}\right\|}_{2}^{2}$ and keep more collaborative reconstruction information in the feature projection space. In addition, CRP uses the local divergence and global divergence of the sample to obtain the discriminativeness of the extracted features, and in our method, we use the label information of the sample to keep the sample in the projection space with a small within-class scatter and larger between-class scatter, thereby improving the discriminativeness of the extracted features.

### 3.2. Algorithmic Procedures

For simplicity, we summarize the algorithmic procedures of the proposed MECRDP as follows:

Step 1:Project the original high-dimensional sample into an intermediate subspace to remove noise and useless information by PCA, and get the projection matrix $P_{V}$. For simplicity, we still utilize X to represent the training samples after the projection.Step 2:Find the eigenvector $x_{v}$ corresponding to the smallest non-zero eigenvalue of $XX^{T}$, then the expanded sample matrix is $\tilde{X}=[X,x_{v}]$.Step 3:Compute the collaborative representation coefficients $\tilde{R}$ by (15) and then reconstruct the samples matrix $\hat{X}$ by $\hat{X}=X\tilde{R}$.Step 4:Compute the collaborative within-class scatter matrix ${\overset{⌢}{S}}_{w}$, the between-class scatter matrix $S_{b}$ and the collaborative reconstructed between-class scatter matrix ${\hat{S}}_{b}$ by (19), (5) and (20), respectively.Step 5:Perform generalized eigenvalue decomposition by ${\overset{⌢}{S}}_{w}p_{i}=\lambda_{i}{\tilde{S}}_{b}p_{i}$, and use eigenvectors corresponding to the smallest d eigenvalues to construct the projection matrix $P=[p_{1},p_{2},\ldots ,p_{d}]$.Step 6:For any input sample $x\in {ℜ}^{n}$, its low-dimensional projection is $y=P^{T}P_{V}^{T}x$.

It is obvious that the proposed method has no iterative steps and its optimal projection matrix can be analytically obtained. Without considering the first step, which is the data preprocessing process that most linear DR methods need to perform, we simply analyze the computational complexity of our proposed method. In step 2, it costs about $O(m^{3})$ to find the eigenvector corresponding to the smallest non-zero eigenvalue of $XX^{T}$, step 3 costs about $O(mn^{2}+n^{3})$, and step 4 needs about $O(mn^{2}+m^{2}n)$, finding the optimal projection matrix in step 5 requires $O(n^{3})$, with the total computational complexity being about $O(m^{3}+m^{2}n+mn^{2}+n^{3})$.

## 4. Experiments

In this section, some experiments are conducted to evaluate the performance of the proposed MECRDP. We compare the recognition performance of these methods on four public databases, including an image object database COIL20 [33], and two face databases ORL [34], FERET [35], and Isolet [36].

### 4.1. Preprocessing and Parameter Setting

In our experiments, all the images were converted to grayscale images and were resized. In the following experiments, each sample is stacked into a column vector in column order and is normalized. Besides, to avoid the singular problem caused by small sample size problems, we reduce the dimension by remaining 98% data energy by PCA. Each database is randomly divided into a training set and test set, say randomly selecting s-samples from each class to form a training set, and the remaining samples are used as a test set. For simplicity, without additional explanation, the nearest neighbor method is used to classify the test samples. Considering that most methods compared have adjustable parameters, for the sake of fairness, we look for the satisfactory parameters in a larger range for them. For example, we find the neighbor parameter for LLDE from 1 to s−1, with and empirically set the rescaling coefficient to 1. In MFA, the parameter for the intrinsic graph is empirically set as s−1 and the parameter for the penalty graph is chosen from {1C,3C,5C,7C,9C}.

Particularly, each experiment was independently repeated 20 times to avoid the bias caused by random selection, and the average results are recorded and reported. All the experiments are implemented in MATLAB R2011b installed on a personal computer (Intel Core i5-4590, 3.30 GHz, 8 Gb RAM).

### 4.2. Experimental Results and Analysis

#### 4.2.1. Experimental on COIL20 Database

The COIL20 [33] is an object recognition database which has 1440 images from 20 objects with 72 images in each object. We resize each image into 32×32 and randomly select 4, 6, 8, 10 images from each class to form the training sample. The parameters λ and α in MECRDP are set to 0.5 and 0.1, respectively.

Table 1 records the maximum average recognition accuracy, the standard derivations, the corresponding dimension, and the average running time of each method on COIL20. Figure 1 shows the average recognition accuracy versus the subspace dimensions on COIL20. From the results in Table 1 and Figure 1, we observe that the proposed MECRDP achieves better recognition accuracy than other methods on COIL20 database. Compared with other methods, the maximum average recognition accuracy of MECRDP is improved about 3%. Table 1 shows the running time of our method is almost the same as other methods, which implies the efficiency of our method. In addition, the optimal recognition result of MECRDP is usually obtained in a lower projection dimension than other methods. The same result could be verified more intuitively in Figure 1. The result in Figure 1 shows that when the dimension of the projection subspace is low, MECRDP is significantly better than other methods. This result means that the features extracted by our method have obvious discriminativeness even in the low-dimensional space. Table 2 shows the maximum average recognition accuracy with different dimension by using nearest neighbor classifier. To evaluate the effectiveness of MECRDP further, we compare it with other dimension reduction methods under another classifier two-layer neural network; the results are shown in Table 3. Table 2 and Table 3 show that the recognition accuracy achieved by the neural network is not as good as that of the nearest neighbor method, but the proposed method always achieves the best results. The result of COIL20 verifies the effectiveness of our method for feature extraction.

#### 4.2.2. Experiment on the ORL Database

The ORL [34] is the face database and consists of 400 images from 40 individuals with 10 images for each individual. These images are varying in lighting, facial expressions, and details. In our experiments, we cropped and resized each image to 32×32, and randomly selected 3, 4, 5, 6 images from each class to form the training set. We set λ and α in MECRDP to 0.05 and 0.9, respectively.

Table 4 reports the maximum average recognition accuracy, the standard derivations, the corresponding dimension, and the average running time of each method on the ORL database. Table 5 and Table 6 list the average recognition rates obtained by the nearest neighbor classifier and neural network classifier for each method, respectively. Figure 2 plots the average recognition accuracy versus the subspace dimensions on ORL. The results show that our method is superior to other methods, especially when the number of training samples is small; the performance improvement of our method is more obvious. For example, when the training samples of each class are 3 and 4, the maximum average recognition accuracy of MECRDP is improved by about 4.1% and 3.1%, respectively. Besides, we note that, for each method, the recognition accuracy is increased as the training sample increases, while that of MECRDP always outperforms other methods. Figure 2 shows that as the projection dimension increases, the performance of some methods, such as LDA, LLDE, and RLSDP, will decrease because they can extract some noise information when the projection dimension is large. However, our method has shown a certain degree of robustness. Comparing Table 5 and Table 6, we find that the recognition accuracy obtained by the nearest neighbor classifier is better than that of the neural network classifier. What’s more, the proposed MECRDP can achieve the best results in almost all the selected feature dimensions. The experimental results on the ORL database verify the advantages of the proposed algorithm.

#### 4.2.3. Experiment on the FERET Database

The FERET [35] is the face database and consists of 13,539 images from 1565 individuals, where we select a subset containing 1400 images from 200 individuals, with seven images for each individual. These images are varying in facial expression, illumination, and pose. We cropped and resized each image to 40×40, and randomly selected 3, 4, 5 images from each class to form the training set. We set λ and α in MECRDP to 0.1 and 0.1, respectively.

Table 7 shows the maximum average recognition accuracy, the standard derivations, the corresponding dimensions, and the average running time of each method on the FERET database. The average recognition rates obtained by the nearest neighbor classifier and neural network classifier for each method are shown in Table 8 and Table 9, respectively. Figure 3 illustrates the average recognition accuracy versus the subspace dimensions on FERET. The results in Table 3 and Figure 3 show that the proposed MECRDP outperforms other methods. It is worth noting that FERET has 200 classes but only seven samples for each class. On the FERET database, the average recognition accuracy is greatly improved by MECRDP. For example, the training samples of each class are 3 and 4, the maximum average recognition accuracy of MECRDP is improved by about 26.2% and 14.1%, respectively. Besides, Figure 3 shows that when the projection dimension increases, the recognition accuracy of MECRDP will decrease but it is still higher than other methods. The results in Table 8 and Table 9 show that the nearest neighbor classifier could achieve higher recognition accuracy than the neural network classifier. Table 8 shows that MECRDP has achieved the highest recognition accuracy in each selected feature dimension. Table 8 shows that MECRDP has achieved the highest recognition accuracy in each selected feature dimension, while Table 9 shows that MECRDP performs better than other feature extraction methods in most cases when the neural network classifiers are used. In general, the experimental results show that the features extracted by MECRDP are more discriminative in most cases.

#### 4.2.4. Experiment on the Isolet Database

The Isolet database [36] was generated as follows. One hundred fifty subjects spoke the name of each letter of the alphabet twice. Hence, there were 52 training examples from each speaker and the size of each sample was 617 × 1. The speakers were grouped into sets of 30 speakers each, and were referred to as isolet1, isolet2, isolet3, isolet4, and isolet5. In our experiment, we compared the performance of each feature extraction method in Isolet1. We set λ and α in MECRDP to 1 and 0.1, respectively.

Table 10 shows the maximum average recognition accuracy, the standard derivations, the corresponding dimensions, and the average running time of each method on the Isolet1 database. Table 11 and Table 12 report the average recognition rates obtained by the nearest neighbor classifier and neural network classifier for each method, respectively. Figure 4 illustrates the average recognition accuracy versus the subspace dimensions in Isolet1. The results in Table 10 show that under different numbers of training samples, the best results are always obtained by MECRDP. Figure 4 shows that as the number of training samples increases, the recognition accuracy of all the feature extraction algorithms is improved. Although the advantages of the proposed MECRDP over other methods will gradually decrease, it can maintain a high recognition accuracy. For example, when the number of training samples for each class is five, the recognition accuracy of MECRDP is improved by at least 3.4% compared to other methods. When the number of training samples for each class is increased to 20, the performance of our method is improved by only 1.2%, and its best recognition accuracy is 94.05%. Table 11 and Table 12 show that MECRDP performs better than other feature extraction methods in most cases. Based on the above experimental results, we believe that the proposed MECRDP is an effective feature extraction method.

#### 4.2.5. Parameter Sensitivity Analysis

Here we analyze the influence of parameters on the proposed algorithm through experiments on the four databases.

Figure 5 and Figure 6 plot the maximum average recognition accuracy of MECRDP versus the parameter λ and α for the four samples of each database, respectively. Figure 4 and Figure 5 show that the parameter selection of MECRDP has a great relationship with the sample database. Within a certain range, such as λ ∊ [0, 1], there is a little change in the recognition accuracy of the four databases. While α has a greater impact on the recognition accuracy for COIL20, ORL, and FERET, particularly for Isolet1, the recognition accuracy of MECRDP was robust for parameter α.

#### 4.2.6. Visualization

In order to compare the distribution of extracted features in low-dimensional space more intuitively, we randomly selected six classes of samples from the ORL database and then projected them into two-dimensional space. Figure 7 shows the projection results of all methods in two dimensions. The results show that most methods have good clustering results except CRP and PCA, which are unsupervised feature extraction methods. This shows that the label information helps to improve the discriminative feature extraction. However, in some classes (class 1 and class 6), some methods, such as CRRP, LDA, MFA, and RLSDP, have not achieved a large between-class distance. Figure 7 shows that the features extracted by MECRDP not only have a small within-class compactness, but also have a larger between-class distance.

In a word, these experimental results imply that using the minimum eigenvector to reduce the sample reconstruction error, and maintaining the collaborative representation relationship between the samples helps to improve the discriminability of the extracted features, especially when the number of training samples is small.

## 5. Conclusions

A new linear dimensionality reduction method based on minimum eigenvector collaborative representation discriminant projection has been proposed for feature extraction in this paper, which can be viewed as an extended collaborative representation projection method. This method employs the eigenvector corresponding to the smallest non-zero eigenvalue of the sample covariance matrix to reduce the collaborative representation error. Meanwhile, the collaborative representation relationship of the samples is maintained in the projection subspace. In addition, the between-class scatter of the reconstructed samples is used to improve the robustness of the projection subspace. The experimental results on four public databases demonstrated the superiority of the proposed method in terms of recognition accuracy as compared with other commonly used linear DR methods. What’s more, the experiments show that the proposed method is especially suitable for dealing with small sample size problems, and it can also work well when the number of training samples is large. Thus, we believe that MECRDP is a general algorithm for feature extraction.

Note that the proposed MECRDP is a parameterized method, and its performance will inevitably be affected by the choice of the parameters, which also happens to other parameterized methods such as CRP, MFA, and RLSDP. In addition, sometimes, as the feature dimension increases, the performance of the algorithm will decrease. How to utilize other information of the sample, such as spatial distribution information, local structure information, and high-order statistical information to further improve the performance and robustness of the algorithm, is also an interesting direction for future study.

## Figures and Tables

**Figure 1 sensors-20-04778-f001:**
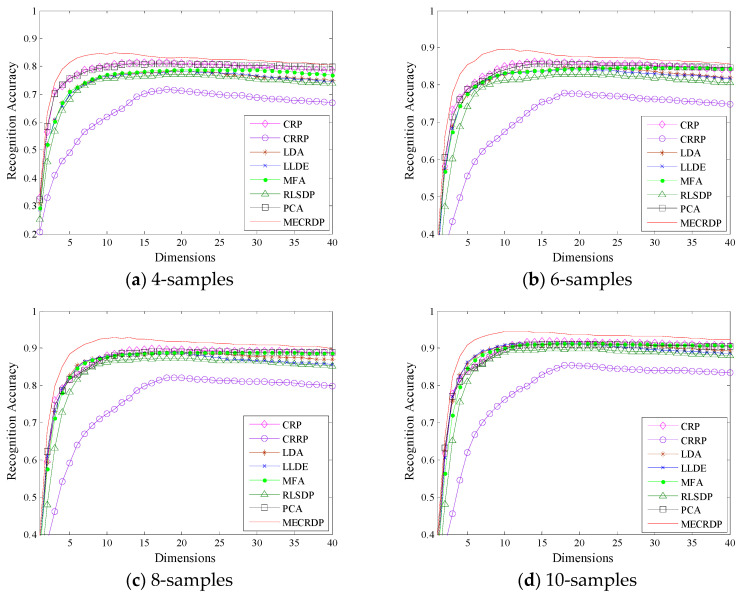
The average recognition accuracy vs. the subspace dimensions of each method in the COIL20 database.

**Figure 2 sensors-20-04778-f002:**
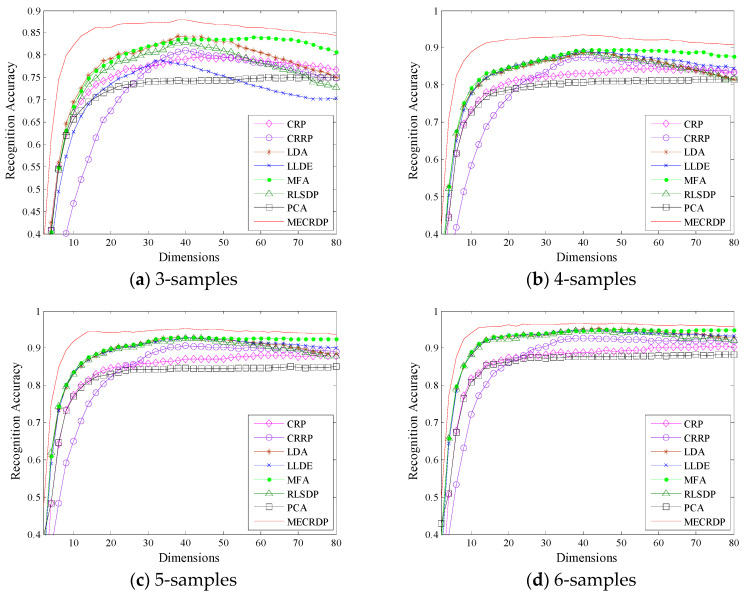
The average recognition accuracy vs. the subspace dimensions of each method in the ORL database.

**Figure 3 sensors-20-04778-f003:**
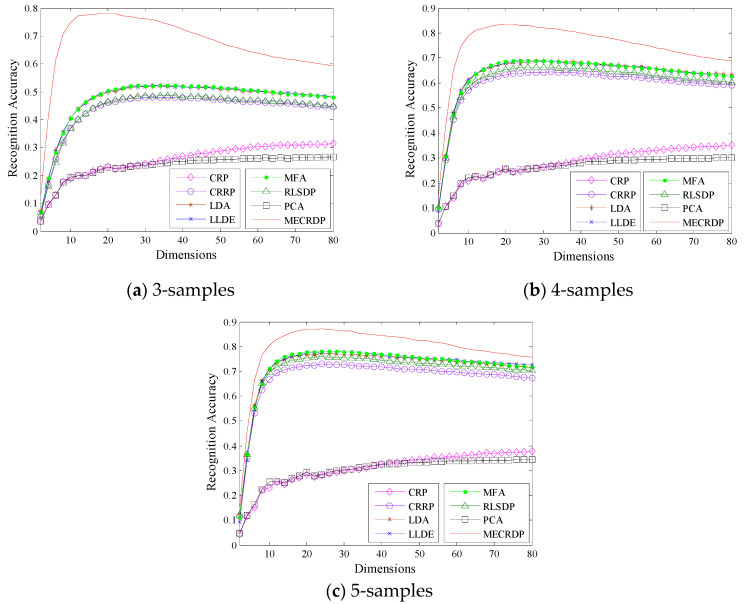
The average recognition accuracy vs. the subspace dimensions of each method in the FERET database.

**Figure 4 sensors-20-04778-f004:**
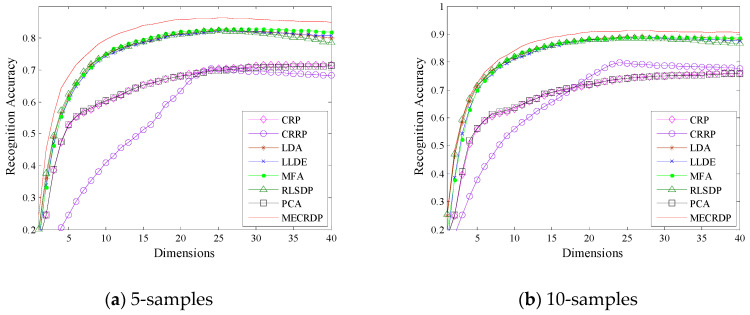
The average recognition accuracy vs. the subspace dimensions of each method in the Isolet1 database.

**Figure 5 sensors-20-04778-f005:**
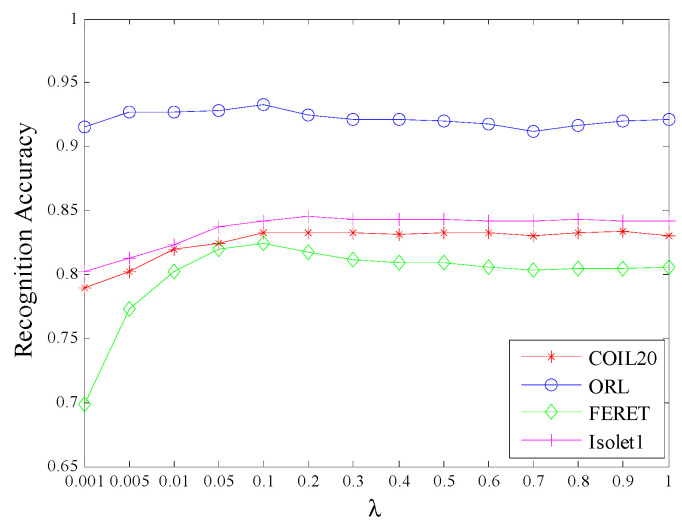
The maximal average recognition accuracy of MECRDP vs. the regularization parameter λ in the four databases.

**Figure 6 sensors-20-04778-f006:**
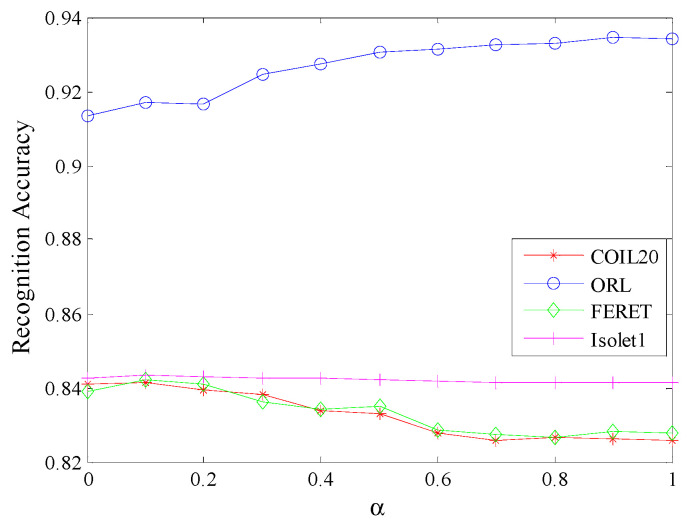
The maximal average recognition accuracy of MECRDP vs. the balance factor α in the four databases.

**Figure 7 sensors-20-04778-f007:**
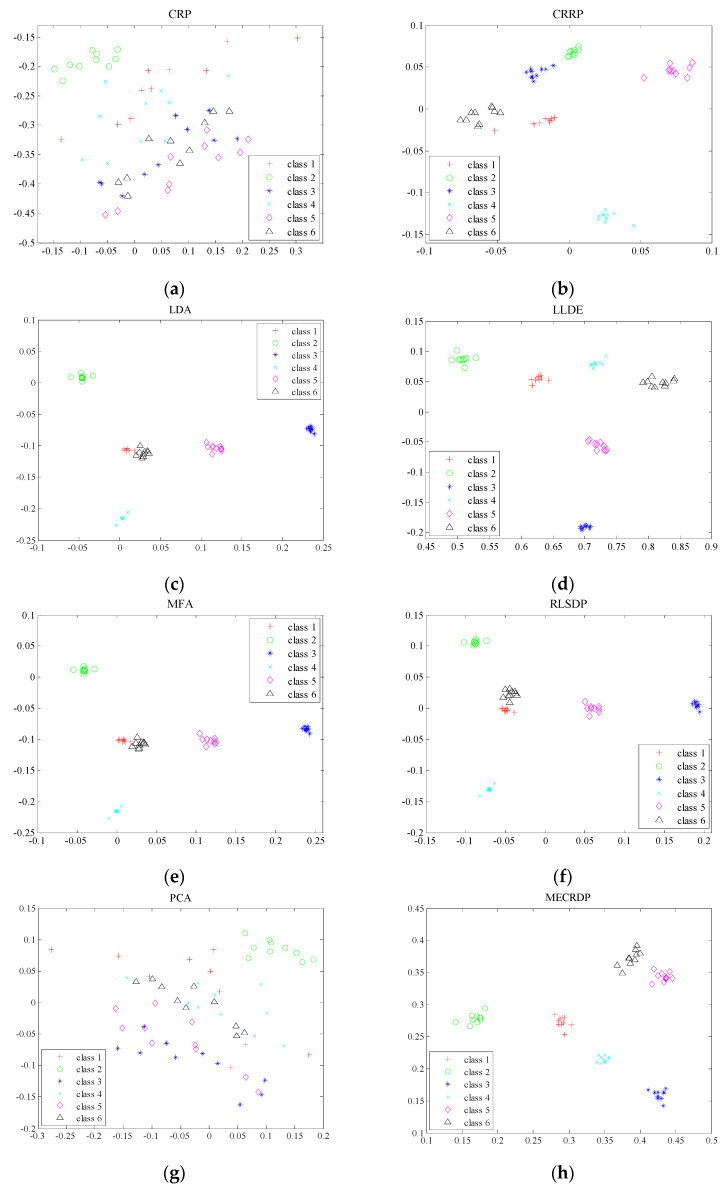
The visualization of six individuals in the ORL database. (**a**) CRP, (**b**) CRRP, (**c**) LDA, (**d**) LLDE, (**e**) MFA, (**f**) RLSDP, (**g**) PCA, (**h**) MECRDP.

**Table 1 sensors-20-04778-t001:** The maximum average recognition accuracy (%) ± the standard derivations (%), the corresponding dimension, and the average running time (seconds) in parentheses of each method in the COIL20 database.

Methods	4-Samples	6-Samples	8-Samples	10-Samples
CRP	81.51 ± 2.08 (16, 0.008)	85.35 ± 2.12 (14, 0.011)	89.90 ± 1.07 (16, 0.017)	91.84 ± 1.39 (17, 0.025)
CRRP	71.70 ± 2.90 (18, 0.007)	77.75 ± 2.59 (18, 0.016)	82.12 ± 2.04 (19, 0.026)	85.47 ± 1.54 (19, 0.042)
LDA	78.36 ± 2.93 (19, 0.008)	84.70 ± 2.28 (22, 0.008)	88.73 ± 1.74 (22, 0.016)	91.30 ± 1.55 (19, 0.017)
LLDE	78.22 ± 3.16 (20, 0.009)	84.27 ± 2.39 (19, 0.015)	88.78 ± 1.73 (19, 0.019)	91.38 ± 1.59 (18, 0.029)
MFA	78.84 ± 2.86 (26, 0.005)	84.65 ± 2.13 (32, 0.010)	88.95 ± 1.80 (21, 0.017)	91.33 ± 1.35 (18, 0.023)
RLSDP	77.25 ± 3.32 (19, 0.005)	82.91 ± 2.50 (18, 0.010)	87.30 ± 1.85 (20, 0.013)	89.83 ± 1.73 (16, 0.020)
PCA	80.84 ± 1.75 (18, 0.004)	85.67 ± 1.68 (14, 0.006)	89.10 ± 1.38 (23, 0.012)	91.14 ± 1.25 (18, 0.016)
MECRDP	84.89 ± 2.26 (11, 0.007)	89.60 ± 2.00 (11, 0.010)	92.79 ± 1.36 (13, 0.018)	94.50 ± 1.07 (13, 0.026)

**Table 2 sensors-20-04778-t002:** The maximum average recognition accuracy (%) in the COIL20 database with different dimension by using nearest neighbor classifier.

Dimension	5	10	15	20	25	30	35	40
CRP	84.31	90.13	91.82	91.75	91.52	91.38	91.05	90.72
CRRP	62.02	76.25	82.75	85.29	84.48	84.15	83.83	83.41
LDA	85.83	90.47	90.92	91.27	91.07	90.59	89.80	89.22
LLDE	86.27	90.80	91.14	91.19	90.45	89.65	89.07	88.61
MFA	84.73	90.10	91.13	91.28	91.06	90.96	90.88	90.73
RLSDP	81.14	88.89	89.76	89.77	89.23	89.00	88.38	87.97
PCA	83.90	89.66	90.98	91.02	90.99	90.76	90.61	90.46
MECRDP	90.80	94.40	94.27	93.71	93.42	93.20	93.69	92.20

**Table 3 sensors-20-04778-t003:** The maximum average recognition accuracy (%) on the COIL20 database with different dimension by using neural network classifier.

Dimension	5	10	15	20	25	30	35	40
CRP	77.21	87.16	89.63	89.12	88.15	87.48	86.37	86.05
CRRP	60.76	74.60	79.71	80.67	80.57	79.98	79.27	78.78
LDA	84.46	87.19	85.19	83.10	82.73	82.65	81.71	81.84
LLDE	85.48	87.75	85.13	83.51	82.62	82.66	82.57	80.67
MFA	83.62	87.52	86.19	83.40	82.98	82.30	82.00	81.44
RLSDP	79.93	85.79	84.63	81.58	81.85	81.88	81.57	81.02
PCA	77.71	86.70	89.85	89.38	88.25	87.66	86.58	85.83
MECRDP	88.77	91.67	91.36	90.85	89.78	89.14	88.03	87.49

**Table 4 sensors-20-04778-t004:** The maximum average recognition accuracy (%) ± the standard derivations (%), the corresponding dimension, and the average running time (seconds) in parentheses of each method in the ORL database.

Methods	3-Samples	4-Samples	5-Samples	6-Samples
CRP	79.55 ± 3.12 (42, 0.019)	84.48 ± 2.32 (52, 0.024)	88.08 ± 2.17 (62, 0.044)	90.56 ± 3.06 (72, 0.058)
CRRP	81.02 ± 2.79 (40, 0.023)	87.46 ± 2.25 (40, 0.034)	90.58 ± 1.85 (40, 0.074)	92.75 ± 1.71 (38, 0.085)
LDA	84.20 ± 2.87 (38, 0.013)	88.75 ± 4.18 (42, 0.018)	92.95 ± 1.85 (42, 0.030)	95.25 ± 1.68 (44, 0.035)
LLDE	78.66 ± 3.26 (34, 0.016)	89.23 ± 2.00 (40, 0.025)	92.80 ± 1.57 (40, 0.045)	94.97 ± 1.62 (44, 0.054)
MFA	84.07 ± 3.19 (58, 0.018)	89.50 ± 1.43 (48, 0.027)	92.90 ± 1.55 (42, 0.034)	95.03 ± 1.17 (48, 0.045)
RLSDP	82.86 ± 2.86 (40, 0.012)	88.94 ± 2.34 (40, 0.018)	92.78 ± 1.57 (40, 0.031)	94.62 ± 1.45 (38, 0.038)
PCA	75.07 ± 2.95 (80, 0.008)	81.44 ± 2.63 (74, 0.012)	85.00 ± 2.81 (68, 0.027)	88.22 ± 2.54 (78, 0.024)
MECRDP	87.92 ± 2.26 (38, 0.014)	93.43 ± 1.62 (40, 0.023)	95.28 ± 1.70 (40, 0.037)	96.66 ± 1.42 (38, 0.054)

**Table 5 sensors-20-04778-t005:** The maximum average recognition accuracy (%) in the ORL database with different dimensions by using the nearest neighbor classifier.

Dimension	10	20	30	40	50	60	70	80
CRP	81.78	87.53	88.25	88.87	89.19	90.16	90.28	90.25
CRRP	72.16	86.03	90.56	92.69	92.28	92.00	91.97	91.84
LDA	88.41	93.12	93.75	95.03	95.00	94.50	93.84	92.81
LLDE	88.28	93.63	93.84	94.94	94.34	93.97	93.75	93.25
MFA	89.00	93.50	94.06	94.75	94.97	94.81	94.81	94.91
RLSDP	88.34	92.56	93.50	94.41	94.12	94.09	92.81	92.31
PCA	80.91	86.19	87.22	87.69	87.87	88.00	88.19	88.19
MECRDP	94.16	96.25	96.25	96.62	96.59	96.06	96.00	95.91

**Table 6 sensors-20-04778-t006:** The maximum average recognition accuracy (%) in the ORL database with different dimensions by using the neural network.

Dimension	10	20	30	40	50	60	70	80
CRP	81.47	88.09	86.72	84.88	83.13	80.56	76.69	75.94
CRRP	71.00	81.28	80.87	78.13	77.53	76.19	72.47	73.56
LDA	85.72	87.53	86.06	78.28	77.69	76.38	75.28	74.69
LLDE	85.97	88.78	84.31	78.09	78.13	77.44	74.94	73.44
MFA	86.97	88.06	84.66	78.97	76.12	73.19	70.88	72.34
RLSDP	85.62	87.38	83.53	79.03	78.31	76.09	74.97	74.31
PCA	80.59	88.44	86.94	84.38	82.12	81.09	79.75	78.22
MECRDP	92.03	90.50	88.39	86.15	84.11	82.28	79.63	78.47

**Table 7 sensors-20-04778-t007:** The maximum average recognition accuracy (%) ± the standard derivations (%), the corresponding dimension, and the average running time (seconds) in parentheses of each method in the FERET database.

Methods	3-Samples	4-Samples	5-Samples
CRP	31.45 ± 1.40 (80, 0.561)	34.99 ± 1.63 (80, 1.189)	37.85 ± 2.67 (80, 2.005)
CRRP	48.08 ± 1.61 (34, 1.362)	64.32 ± 1.69 (32, 2.479)	72.91 ± 1.31 (24, 3.924)
LDA	52.03 ± 1.82 (34, 0.443)	68.64 ± 1.92 (32, 0.809)	77.30 ± 1.82 (26, 1.314)
LLDE	52.32 ± 1.86 (34, 0.551)	68.89 ± 1.61 (26, 1.029)	77.84 ± 1.89 (24, 1.630)
MFA	52.28 ± 2.04 (36, 0.454)	68.95 ± 1.99 (22, 0.864)	78.19 ± 1.57 (24, 1.387)
RLSDP	48.57 ± 1.71 (36, 0.480)	66.35 ± 1.54 (26, 0.873)	76.01 ± 1.27 (24, 1.374)
PCA	26.52 ± 0.99 (80, 0.245)	30.15 ± 1.11 (80, 0.460)	34.45 ± 2.03 (80, 0.806)
MECRDP	78.13 ± 1.71 (18, 0.563)	83.66 ± 1.12 (20, 1.038)	87.00 ± 1.26 (24, 1.727)

**Table 8 sensors-20-04778-t008:** The maximum average recognition accuracy (%) in the FERET database with different dimensions by using the nearest neighbor classifier.

Dimension	10	20	30	40	50	60	70	80
CRP	23.18	28.36	30.01	32.56	34.40	35.70	36.99	37.85
CRRP	66.75	74.49	72.58	71.66	70.66	69.55	68.65	67.42
LDA	70.65	76.89	76.99	76.05	75.10	73.93	72.85	71.42
LLDE	71.09	77.57	77.80	76.56	75.36	74.69	73.50	72.55
MFA	71.10	77.71	77.74	77.05	75.44	74.13	73.14	71.73
RLSDP	69.68	75.43	75.53	74.51	73.30	72.14	71.39	70.28
PCA	25.35	29.21	30.16	32.56	33.39	33.76	34.21	34.45
MECRDP	80.55	86.89	86.11	84.55	82.42	79.81	77.56	75.65

**Table 9 sensors-20-04778-t009:** The maximum average recognition accuracy (%) in the FERET database with different dimensions by using the neural network.

Dimension	10	20	30	40	50	60	70	80
CRP	17.06	35.78	42.63	46.09	47.26	45.73	43.16	41.14
CRRP	51.83	58.00	52.88	47.17	42.04	37.91	34.28	31.39
LDA	55.03	61.86	56.45	50.16	45.00	40.05	36.50	33.14
LLDE	54.88	61.86	56.11	50.15	45.22	40.37	35.74	32.75
MFA	56.83	63.69	57.98	51.37	45.81	40.21	35.33	32.76
RLSDP	53.85	59.84	54.48	48.88	43.14	39.68	35.45	32.90
PCA	17.65	35.99	44.18	47.09	47.51	46.80	45.05	43.19
MECRDP	68.91	78.88	72.30	65.06	57.85	50.26	44.90	42.92

**Table 10 sensors-20-04778-t010:** The maximum average recognition accuracy (%) ± the standard derivations (%), the corresponding dimensions, and the average running time (seconds) in parentheses of each method in the Isolet1 database.

Methods	5-Samples	10-Samples	15-Samples	20-Samples
CRP	71.59 ± 1.24 (35, 0.015)	76.11 ± 1.39 (39, 0.054)	78.80 ± 0.78 (40, 0.125)	80.72 ± 1.01 (40, 0.209)
CRRP	70.47 ± 2.25 (24, 0.025)	79.59 ± 1.21 (24, 0.073)	83.79 ± 1.17 (25, 0.223)	86.25 ± 1.10 (24, 0.412)
LDA	82.59 ± 1.87 (25, 0.012)	89.12 ± 1.08 (25, 0.030)	91.55 ± 0.76 (25, 0.064)	92.80 ± 0.68 (24, 0.118)
LLDE	82.48 ± 1.89 (26, 0.016)	88.89 ± 0.94 (26, 0.055)	91.65 ± 0.75 (26, 0.118)	92.86 ± 0.59 (26, 0.217)
MFA	82.94 ± 1.72 (29, 0.013)	88.93 ± 0.98 (25, 0.043)	91.59 ± 0.87 (25, 0.099)	92.71 ± 0.82 (24, 0.169)
RLSDP	82.31 ± 1.72 (25, 0.011)	88.71 ± 0.97 (26, 0.034)	91.42 ± 0.53 (25, 0.077)	92.80 ± 0.67 (25, 0.144)
PCA	71.29 ± 1.57 (40, 0.007)	75.86 ± 1.45 (40, 0.022)	79.09 ± 0.87 (39, 0.049)	80.91 ± 1.13 (40, 0.089)
MECRDP	86.40 ± 1.61 (25, 0.013)	91.36 ± 1.05 (26, 0.046)	93.23 ± 0.77 (25, 0.103)	94.05 ± 0.68 (25, 0.199)

**Table 11 sensors-20-04778-t011:** The maximum average recognition accuracy (%) in the Isolet1 database with different dimensions by using the nearest neighbor classifier.

Dimension	5	10	15	20	25	30	35	40
CRP	59.25	67.74	73.92	76.97	79.15	80.00	80.34	80.72
CRRP	48.04	66.51	75.96	83.81	86.23	86.02	85.75	85.59
LDA	75.03	84.04	89.88	91.83	92.79	92.73	92.64	92.36
LLDE	73.01	82.88	89.40	91.58	92.83	91.69	90.58	89.92
MFA	72.20	85.12	90.28	92.03	92.68	92.37	92.24	92.33
RLSDP	75.94	84.21	89.91	91.75	92.80	92.37	92.14	91.63
PCA	58.51	67.83	74.08	77.25	79.32	80.04	80.23	80.91
MECRDP	74.72	86.62	91.82	93.26	94.05	93.82	93.73	93.57

**Table 12 sensors-20-04778-t012:** The maximum average recognition accuracy (%) in the Isolet1 database with different dimensions by using the neural network classifier.

Dimension	5	10	15	20	25	30	35	40
CRP	63.29	74.57	83.81	85.74	87.16	87.41	87.55	87.13
CRRP	50.40	66.66	75.35	80.59	81.92	81.40	80.44	79.51
LDA	73.79	81.86	85.65	86.40	85.50	84.89	84.07	83.75
LLDE	70.66	80.75	85.86	86.70	86.14	85.39	85.08	84.30
MFA	71.56	82.97	86.58	87.63	86.38	84.95	84.04	83.00
RLSDP	74.24	81.78	85.86	86.29	85.90	85.10	84.17	83.53
PCA	63.65	75.17	84.34	86.29	87.08	87.81	87.75	87.83
MECRDP	73.49	85.08	89.17	88.94	89.54	89.19	89.23	89.06

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
