# Peer review of "Minimum Eigenvector Collaborative Representation Discriminant Projection for Feature Extraction"

_sensors, 2020, doi:10.3390/s20174778_

Round 1

Reviewer 1 Report

The authors proposed a novel method of dimensionality reduction. This topic is still very popular in the field of signal processing and the results demonstrated the effectiveness of the proposed approach. The paper is well written, however, due to limited time for this review I was not able to verify the correctness of the provided mathematical formulas. For the current manuscript, I have the following comments.

  1. The article contains some minor spelling errors, which should be corrected e.g. “For exmples,” → “For examples,” in Line 253.

  2. The authors selected some “less challenging” datasets designed for image recognition problems. This does not allow to evaluate if the proposed method is also robust against outliers and is beneficial for other problems than image recognition. LFW Face datasets (http://vis-www.cs.umass.edu/lfw/), CIFAR dataset (https://www.cs.toronto.edu/~kriz/cifar.html) or another. The authors could also consider to evaluate their approach for other computer vision problems than image recognition or for other signal processing problems. This will help to show that the proposed method can be applied for more challenging data. Therefore, I suggest to extend this article by adding at least one additional dataset.

  3. In Section “Results”, the authors provide the information how many features were selected only for COIL20 dataset. This information is missing for other datasets. The authors also do not discuss how many features were selected for reference methods in the table. In this kind of work, it is common practice to show the results depending on how many features were selected for all methods. Currently, this information is only presented on the plots, but missing in the text and therefore it is hard to establish for how many features the tables were generated e.g. the table presents the accuracy for the same number of features?

  4. In the results section, the authors could also present as the reference, the results for reference methods if they were trained on a larger number of samples to show if method is only better if we have small number of training samples or it could also work well if we more training samples is available. Now it is not clear if the proposed method will be beneficial only if a small number of training samples is available or it is also comparable if more samples are available.

  5. I would like to see also comparison the classical PCA method as it is a relatively simple and still widely used method.

  6. The presented results were shown only for the nearest neighbor method what is a significant drawback of this paper. The authors should also extend the results using at least one more robust classifier.

  7. The legends In Fig. 6 shall be corrected because a part of the text is outside the legend box.

  8. In Fig. 3 I suggest moving the legend outsize the presented plots as one row legend.

  9. In Section 4.2.1. wrote that “Table 1 shows the running time of our method is almost the same as other method”, however, Tab. 1 in the manuscript presents “The maximum average recognition accuracy”. So, the table with the running time summary is missing. You should also provide how the reference methods were implemented, by implemented by the authors of this manuscript or using the codes provided by the original authors?

Author Response

Thank you for your valuable comments. We carefully reviewed and revised the manuscript based on your comments. The revised parts in the manuscript are highlighted in red. The detailed response can be seen in the attachment.

Reviewer 2 Report

In this paper, a minimum eigenvector collaborative representation discriminant projection method is proposed to extract the features with strong discriminant. The authors firstly use the eigenvector corresponding to the smallest non-zero eigenvalue of the sample covariance matrix to reduce the collaborative representation error of each sample. Then, they maintain the collaborative representation relationship of the samples to improve the discriminability of the extracted features. The results demonstrate that the proposed method outperforms other dimensionality reduction methods in terms of recognition accuracy. The manuscript is well written. The experiment demonstrates the effectiveness of the proposed method. So, I suggest this manuscript be accepted.

Author Response

Thank you for your valuable comments. We carefully reviewed and revised the manuscript based on your comments.

Reviewer 3 Report

The article presents a new method to reduce the dimensionality in problems whose number of samples is less than the dimensionality number of the problem. The article presents tests in three public databases (COIL20, ROL, and FERET), and in all of them, it has demonstrated superior performance. The article is well written, there are only a few typing errors (line 79 and 90). Some important points to consider for a review of the article are:

1. Improve the explanation of the effect of including xv in the X~ matrix (section 3.1). Why it improves the stability of the proposed technique versus the CRP algorithm?

2. Describe the procedure of the algorithm (section 3.2) in a more computational way and share the proposed technique in some software tools (Python, R, Matlab).

2. It is not clear what is an appropriate alpha value that explains the performance differences between the COIL20 and ENT databases as the parameter increases (section) 4.2.4.

3. Conclusions are weak, cases of algorithm failure, future improvements and the level of numerical stability are not adequately indicated

Author Response

Thank you for your valuable comments. We carefully reviewed and revised the manuscript based on your comments. The revised parts in the manuscript are highlighted in red. A detailed response is provided in the attachment.

Round 2

Reviewer 1 Report

The article was significantly improved in comparison to the previous version and and can be published in the Sensors journal.